# The Growth-Promoting and Colonization of the Pine Endophytic *Pseudomonas abietaniphila* for Pine Wilt Disease Control

**DOI:** 10.3390/microorganisms12061089

**Published:** 2024-05-27

**Authors:** Yueyuan Peng, Yuwei Tang, Da Li, Jianren Ye

**Affiliations:** 1Co-Innovation Center for Sustainable Forestry in Southern China, College of Forestry, Nanjing Forestry University, Nanjing 210037, China; benben@njfu.edu.cn (Y.P.); tangyuwei96@outlook.com (Y.T.); ld18851862709@163.com (D.L.); 2Jiangsu Key Laboratory for Prevention and Management of Invasive Species, Nanjing Forestry University, Nanjing 210037, China

**Keywords:** pine wilt disease, *Pseudomonas*, *Pinus massoniana*, growth-promoting, disease prevention, bacterial colonization

## Abstract

In this study, we focused on evaluating the impact of *Pseudomonas abietaniphila* BHJ04 on the growth of *Pinus massoniana* seedlings and its biocontrol efficacy against pine wilt disease (PWD). Additionally, the colonization dynamics of *P. abietaniphila* BHJ04 on *P. massoniana* were examined. The growth promotion experiment showed that *P. abietaniphila* BHJ04 significantly promoted the growth of the branches and roots of *P. massoniana*. Pot control experiments indicated that strain BHJ04 significantly inhibited the spread of PWD. There were significant changes in the expression of several genes related to pine wood nematode defense in *P. massoniana*, including chitinase, nicotinamide synthetase, and triangular tetrapeptide-like superfamily protein isoform 9. Furthermore, our results revealed significant upregulation of genes associated with the water stress response (dehydration-responsive proteins), genetic material replication (DNA/RNA polymerase superfamily proteins), cell wall hydrolase, and detoxification (cytochrome P450 and cytochrome P450 monooxygenase superfamily genes) in the self-regulation of *P. massoniana*. Colonization experiments demonstrated that strain BHJ04 can colonize the roots, shoots, and leaves of *P. massoniana*, and the colonization amount on the leaves was the greatest, reaching 160,000 on the 15th day. However, colonization of the stems lasted longer, with the highest level of colonization observed after 45 d. This study provides a preliminary exploration of the growth-promoting and disease-preventing mechanisms of *P. abietaniphila* BHJ04 and its ability to colonize pines, thus providing a new biocontrol microbial resource for the biological control of plant diseases.

## 1. Introduction

*Pinus massoniana* Lamb., a cornerstone species for fast-growing and high-yield timber and resin forests in China, is part of a group of rapidly growing, drought-tolerant evergreen trees extensively found across subtropical regions of the country. Nevertheless, it is susceptible to pests and diseases, the most serious being pine wilt disease (PWD) [1,2,3]. PWD is a devastating forest disease caused by the pine wood nematode (PWN) *Bursaphelenchus xylophilus*, which is highly pathogenic and spreads rapidly [4]. To date, large forest areas in Japan, the Republic of Korea, China, and southern Europe have been infested by the PWN, and these levels are expected to increase by 50% over the next 50 years [5]. PWD management predominantly involves the removal and destruction of infected or deceased pines through burning or felling [6], the use of traps for vector control, stem injections [7], and the application of chemical treatments [8]. However, these control strategies have great drawbacks, including high costs, limited safety, short-term effectiveness, and considerable environmental impact. Moreover, plant growth-promoting endophytes (PGPEs) have gained attention for their ability to enhance plant growth and bolster resistance against pathogens [9,10,11].

PGPEs’ functions in plants stability has become well known [12]. Microbial activity can directly stimulate plant growth through various mechanisms, including the production of molecules that solubilize inorganic phosphorus and potassium [13], the generation of siderophores [14], nitrogen fixation [15], and the biosynthesis of phytohormones such as gibberellins, cytokinins, and auxin (indole acetic acid) [16]. Additionally, microbes contribute to plant growth by synthesizing alginate, a highly stable, water-structured sugar molecule, and the enzyme 1-aminocyclopropane-1-carboxylate deaminase [17], further supporting plant development [18,19]. Alternatively, it can trigger systemic resistance within the target plant species, thereby enhancing the plant’s resilience to stressors and facilitating sustained growth [16]. Furthermore, plants can produce substances that confer resistance against pathogenic bacteria, including synthetic antibiotics, hydrogen cyanide, hydrolases that degrade fungal cell walls, iron chelators, and volatile organic compounds that inhibit phytopathogens [20]. PGPEs can mitigate the risk of disease in plants and foster robust growth through various mechanisms.

The biocontrol microorganisms of *Pinus* can promote growth, induce resistance, and treat PWD according to published research. For example, the strain *Serratia proteamaculans* BXF1, which was isolated from the PWN and has no virulence effect on pine seedlings, is likely derived from an endophytic bacterial community in the host pine. Strain BXF1 produces siderophores, indole acetic acid (IAA), ethyl acetate and ammonia, which promote the growth of tomatoes and pine and colonize them [21]. When pine seedlings were treated with 1-amino cyclopropane carboxylic acid-producing *Pseudomonas putida* UW4 [17] and its mutants, it was found that the wild-type strain promoted the development of the crown and alleviated the symptoms of PWD in pine seedlings compared to the mutants [22]. Foliar sprays of *Pseudomonas koreensis* IRP7 and *Lysobacter enzymogenes* IRP8 greatly reduced the severity of PWD and increased the relative abundance of beneficial interroot microorganisms in susceptible pines [23]. Furthermore, increasing evidence indicates that chitinase enhances the resistance of pine trees to nematode pathogens [24,25]. Thirteen bacterial strains isolated from pine needles and stems, when used to treat pine trees, exhibited a similar expression of at least four defense-related genes (including chitinase) as pine trees subjected to γ-aminobutyric acid treatment. *P. putida* 16YSM-E48, *Curtobacterium pusillum* 16YSM-P180, and *Stenotrophomonas rhizophila* 16YSM-P39 significantly reduce the severity of PWD in pot experiments [26]. Additionally, chitinase extracted from *Lysobacter capsica* has been shown to decrease hatching speed and induce damage to the eggshells of root-knot nematodes [27]. Treatment of pine trees infected with PWD using the phytoendophyte *Bacillus thuringiensis* JCK-1233 promotes the expression of phytophthora-associated genes (*pr-1*) and defense-associated genes and effectively inhibits the development of PWD, whereas *B. thuringiensis* JCK-1233 has no direct nematicidal activity [28].

In our preliminary research, we isolated a predominant endophyte strain, *Pseudomonas abietaniphila* BHJ04, from the branches of healthy *P. massoniana* pines. Preliminary investigations indicated that this strain can enhance vegetative growth and successfully colonize *P. massoniana* tissues (unpublished). Thus, our aims of this research were threefold: first, to ascertain the growth-promoting impact of the BHJ04 strain on *P. massoniana* seedlings; second, to examine the capacity of BHJ04 to inhibit PWD in *P. massoniana* and to analyze the expression of associated genes; and third, to explore the colonization dynamics of the BHJ04 strain within *P. massoniana*. This study offers a foundational investigation into the mechanisms by which *P. abietaniphila* BHJ04 promotes growth and prevents disease, as well as its capacity to colonize pine species. We identified and characterized this microorganism as a novel biocontrol agent, thereby enriching the repertoire of microbial resources available for the biological management of plant diseases. By elucidating the specific interactions between *P. abietaniphila* BHJ04 and its host plants, this study contributes to a deeper understanding of its potential utility in sustainable agricultural practices. This exploration not only advances our knowledge of microbial interactions in forestry but also underscores the importance of developing effective, environmentally friendly alternatives to chemical pesticides for the control of plant pathogens.

## 2. Materials and Methods

### 2.1. Evaluation of Plant Growth-Promoting Traits of Bacterial Strains In Vitro

The *P. abietaniphila* BHJ04 (strain number: CCTCC AB 2024052) bacterial strains were isolated from healthy stem sections of *P. massoniana* located at the Nanjing Baima Base. The methodology was as follows. Sterilized Haglof 300 mm increment borers (Haglof, Långsele, Sweden) were employed to extract stem samples from *P. massoniana*. After thoroughly disinfecting the surface of the tissue samples, excess moisture was removed with sterile filter paper. The samples were then cut into approximately 0.5 cm segments using sterile scissors. These segments were immersed in an Erlenmeyer flask containing 20 mL of sterile water and shaken at 28 °C and 180 rpm for 30 min. Next, 1 mL of the suspension was mixed with 9 mL of sterile water and serially diluted to concentrations of 10^−1^, 10^−2^, 10^−3^, 10^−4^, and 10^−5^. From each dilution, 100 μL was spread onto NA plates and incubated for 1-3 days. Colonies were selected based on distinctive morphological characteristics, including shape, color, edge, transparency, and moisture, and were purified by streaking to isolate the BHJ04 strain. Its potential for promoting plant growth was evaluated through in vitro screening.

### 2.2. Indole Acetic Acid (IAA) Production

Estimation of IAA in the bacterial fermentation broth was performed using a colorimetric assay. The endophytes were cultured in 20 mL of LB broth at 28 °C and 200 rpm for 12 h in a shaking incubator to obtain the seed liquid. Two hundred microliters of seed liquid was added to 20 mL of LB broth containing 0.1 g/L L-tryptophan. The cells were cultivated in a shaking incubator at 30 °C and 200 rpm. Four days later, 1 mL of bacterial culture was collected and centrifuged at 10,000× *g* for 10 min at 25 °C. Then, 200 μL of the supernatant was mixed with 400 mL of Salksowski II colorimetric solution (0.5 M FeCl_3_, 1 mL + 35% HClO, 450 mL) in a 1.5 EP tube and kept in the dark for 30 min, after which a UV visible spectrophotometer was used to measure the absorbance at 530 nm. The IAA standard curve was prepared by zeroing the standard solution at 0 mg/L and plotting the IAA standard curve with the standard concentration of IAA as the horizontal coordinate and the absorbance value as the vertical coordinate.

### 2.3. Potassium Solubilization

The potassium solubility of the strain was evaluated by using solid Aleksandrov medium [29], which contained (per liter) 5 g of glucose, 0.005 g of MgSO_4_·7H_2_O, 0.1 g of FeCl_3_, 2.0 g of CaCO_3_, 3.0 g of potassium feldspar powder, 2.0 g of calcium phosphate, and 20 g of agar [30]. Ten microliters of fresh bacterial cultures were inoculated into the center of the medium, and the plates were incubated at 30 °C. After 7 days, the size of the zone of clearance increased. The larger the zone of clearance was, the greater the ability to solubilize potassium, and the K solubilization efficiency (KE = diameter of solubilization halo/diameter of the colony) was calculated for K according to Yaghoubi Khanghahi et al. [31].

### 2.4. Phosphate Solubilization

The qualitative test for bacterial phosphorus solubilization was carried out using inorganic phosphorus (IP) medium and organic phosphorus solid medium (OP) solid media plate assays [32]. Ten microliters of activated bacterial solution from the roots and leaves of saline alkaline plants that can cultivate endophytic bacteria were placed in the center of a PKO inorganic medium plate and incubated inverted at 28 °C. After 7 days, the plate was removed, and the zone of clearance that formed around the bacterial colonies indicated the solubilizing effect of inorganic phosphorus. The diameter of the zone of clearance was measured, and the stronger the phosphorus solubilizing ability was, the larger the zone of clearance was. The diameter of the hydrolysis circle was measured.

### 2.5. Siderophore Production

The chromium azurite S (CAS) method was used for the detection of iron carrier production [33]. Ten microliters of fresh bacterial cultures grown in iron-deficient media were inoculated on CAS agar plates and then incubated at 28 °C for 3–4 days. The formation of a zone of clearance around the bacterial colonies indicated the production of iron carriers. Iron-deficient medium without bacteria was used as a negative control.

### 2.6. Nitrogen Fixation

The growth of the strain in Ashby nitrogen-free solid medium was used to determine the preliminary nitrogen fixation ability of the strain. The strain activated with LB culture solution was streaked on Ashby medium, incubated at 28 °C for 3–5 days, and then passaged three times in succession to observe and record the growth of the strain.

### 2.7. Promoting Effect of the Strain on P. massoniana Seedlings

The endophytic bacterial inoculant was prepared by plating the isolated bacteria on LB agar plates and incubating them at 30 °C for 24 h. From the fresh plates, the bacterial cultures were suspended in LB broth and incubated at 200 rpm at 30 °C for 12 h. The endophytic bacterial suspensions were then centrifuged and resuspended in distilled water to a concentration of 10^6^ CFU/mL. Two-year-old, clean-growing pine plants were selected, and six plants were established in each treatment group. Plastic pots (14.7 cm × 11.5 cm × 13.5 cm) were used to colonize 1 plant in each pot under natural light, and 50 mL of bacterial suspension (10^6^ CFU/mL) and LB were inoculated in each treatment.

After two months, the plant height and soil diameter of the potted plants were measured, the whole plant was excavated, the soil attached to the root system was removed, the roots were washed in tap water to keep the root system intact, and the roots were laid flat on filter paper to absorb the water. Then, the fresh mass of the roots, stems, and leaves of the potted plants was weighed, and the plants were subsequently dried in an oven at 115 °C for 4 h to determine the dry weight.

### 2.8. Induction of Resistance of P. massoniana to Pine Wilt Disease by P. abietaniphila BHJ04

The bacteria *P. abietaniphila* BHJ04 was cultured in LB solid media at 30 °C for 48 h. Single colonies were removed from the LB broth and incubated for 12 h at 28 °C and 200 rpm in a shaking incubator, and the bacterial culture concentrations were OD_600_ = 0.8.

The PWN (*B. xylophilus*) used was the highly virulent strain AMA3, which was conserved at the Forest Pathology Laboratory of Nanjing Forestry University. *Botrytis cinerea* was placed on potato dextrose agar and incubated at 25 °C for one week; subsequently, 200 μL of PWN solution was added to the culture. After incubation, the PWN was extracted from the fungal cultures using the Bellman funnel method [34]. The PWN suspension was adjusted to a concentration of 2000 nematodes/mL in sterile distilled water (SDW).

When the bacterial culture reached a concentration of OD_600_ = 0.8, the culture was centrifuged at 4000× *g* for 10 min and then enriched with 1 × phosphate-buffered saline (PBS) until the bacterial concentration reached 8 × 10^8^ CFU/mL. Tween 20 (250 mg/L) was then added to the bacterial cell suspension.

Healthy two-year-old *P. massoniana* seedlings were selected to evaluate the efficacy of BHJ04 against PWD. Pine seedlings grown in a nursery received two 5 mL foliar sprays of a bacterial suspension prepared according to the described method every two weeks. One week after treatment, the PWN aqueous suspension (2000 nematodes/mL) was pipetted into dry cotton and placed in a small cut in the trunk of a pine tree using a surface sterilizer. The inoculated cotton was then covered with a sealing film to prevent it from drying. Control seedlings were exposed to SDW, and four replicates were monitored for each treatment [35]. Regular monitoring and disease severity were evaluated using a 5-degree scale as follows: 0 = healthy seedlings showing no wilting or needle-browning symptoms, 1 = <20% needle browning, 2 = 20–39% needle browning, 3 = 40–59% needle browning, 4 = 60–79% needle browning and terminal shoot bending, and 5 = 80–100% needle browning and wilting of the whole seedling [26]. The observation period was 30 d, during which the condition of the pines was monitored every three days. Images were taken to record the state of the trees on days 0 and 30.

Needles of *P. massoniana* pine from each treatment group were collected at 6 h, 12 h, and 24 h postinoculation with PWNs, flash frozen in liquid nitrogen, and subsequently stored at −80 °C. Total RNA was extracted from the pine needles using a *SteadyPure* Plant RNA Extraction Kit (Accurate Biology, Changsha, China) to eliminate high viscosity and profuse polysaccharides. cDNA libraries were synthesized from total RNA using reverse transcriptase PCR. Relative transcript levels of resistance genes in treated *P. massoniana* needles were detected using quantitative real-time PCR (RT-qPCR). The primer sets included eight defense-related genes, namely, nicotianamine synthase (NAS), chitinase, DNA/RNA polymerase superfamily protein, tetratricopeptide repeat-like superfamily protein isoform 9, dehydration responsive protein, cell wall-associated hydrolase, cytochrome P450 monooxygenase superfamily, and cytochrome P450 [36] (Table 1), with actin serving as the internal reference gene. The relative transcript abundance of defense-related genes was subjected to RT-qPCR using a *SYBR* Green Pro Taq HS premixed qPCR kit with high Rox (Accurate Biology, Changsha, China). Primer specificity was examined with melt curve analysis before starting RT-qPCR. The reaction conditions were 95 °C for 30 s, followed by 40 cycles of 95 °C for 5 s and 60 °C for 30 s. The conditions for the melt curve analysis were 95 °C for 15 s, 60 °C for 60 s, and 95 °C for 30 s.

### 2.9. Construction of the EGFP-Labelled Strain BHJ04 and Colonisation Assay

The plasmid pBBR1MCS2 containing enhanced green fluorescent protein (EGFP) was introduced into *P. abietaniphila* BHJ04 using heat-induced transformation, and green fluorescence-labelled *P. abietaniphila* was screened on LB plates supplemented with kanamycin and visualized using a Zeiss fluorescence microscope.

The *egfp*-labelled strain BHJ04 was incubated in LB liquid media supplemented with kanamycin at 28 °C and 200 rpm for 24 h. Then, the solution was centrifuged at 10,000× *g* for 10 min at 4 °C. The supernatant was discarded, and the precipitate was suspended in PBS buffer (7.4) to produce inoculant at a concentration of approximately 10^8^ CFU/mL. Five milliliters of the inoculum was inoculated into the leaves, stems, and roots of two-year-old *P. massoniana* seedlings via noninvasive spraying, skin grafting and root irrigation, respectively. The seedlings were placed in an outdoor incubator. Leaves, stems, and roots of the pine seedlings in the treatment and control groups were subsequently collected at 0 d, 15 d, 30 d, 45 d, and 60 d after inoculation and stored at −80 °C. Total genomic DNA from each sample was extracted using a plant genomic DNA kit (Accurate Biology, Changsha, China), and the concentration and quality of the extracted genomic DNA were determined using a Nanodrop spectrophotometer. The DNA of each sample was diluted to a concentration of less than 100 ng/μL and stored at −20 °C. 

For fluorescence microscopy, the leaves, stems, and roots of *P. massoniana* were collected at the 30th observation time point. The samples were examined using a Zeiss LSM710 (Carl Zeiss AG, Thuringia, Germany) laser confocal scanning microscope equipped with a 100× objective lens. The excitation wavelength for GFP fluorescence was set at 514 nm, with emission detected within the range of 520–540 nm.

The complete genome information of strain BHJ04 was obtained using a combination of second- and third-generation sequencing methods based on Oxford Nanopore sequencing technology (Grandomics, Wuhan, China). Firstly, DNA from the strain was extracted using the Qiagen kit (Qiagen, Hilden, Germany). Subsequently, single-molecule sequencing was performed using the PromethION sequencer from Oxford Nanopore Technology, generating raw sequencing data. The quality-controlled second- and third-generation sequencing data were hybrid assembled using a Unicycler (version 0.4.8). The assemblies were corrected using Pilon (version 1.23) or NextPolish (version 1.4.13) with second-generation sequencing data. Custom scripts were utilized to check for circularization of the corrected genome and to remove redundant segments. For circularized sequences, the origin was shifted to the replication start point of the genome using Circlator (version 1.5.5), thus obtaining the final genome sequence of strain BHJ04. The whole-genome sequence of BHJ04 has been deposited in National Center for Biotechnology Information website (NCBI, https://www.ncbi.nlm.nih.gov. Accessed on 17 May 2024) and assigned an accession number (CP 155619). 

Collinearity analysis was conducted for the whole genomes of nine *Pseudomonas* species (*P. abietaniphila* BHJ04, *P. koreensis* CP155621, *P. abietaniphila* FNCO01000054., *P. abietaniphila* NZ_BBQJ00000000.1, *P. abietaniphila* NZ_BBQR00000000.1, *P. aeruginosa* NC_002516.2, *P. fluorescens* NZ_LT907842.1, *P. kribbensis* NZ_CP029608.1, and *P. putida* NC_021505.1). Except for strain BHJ04, the whole-genome data for all the other strains mentioned were sourced from the NCBI (Accessed on 20 June 2023) using Mauve software (20150226). The major objectives were to identify and compare homologous sequence blocks across these bacterial genomes, pinpoint the specific gene of BHJ04, and compare its gene fragments against those in the NCBI database to confirm its specificity. 

The bacterial DNA was extracted using the freeze-thaw method. The PCR amplification products of the target genes were cut, collected, and tested at 80 ng/μL. Then, the 10-fold dilution of the product was used as a template for fluorescence quantitative PCR amplification. The validity of the established fluorescence quantitative PCR method was tested based on the R^2^, slope, and amplification efficiency (E) of the standard curve.

Real-time fluorescence quantitative PCR was performed using the extracted total genomic DNA as a template. Four technical replicates were established for each sample. A two-step amplification protocol was used for the amplification reactions. The reaction conditions were 95 °C for 30 s, followed by 40 cycles of 95 °C for 5 s and 60 °C for 30 s. The conditions for the melt curve analysis were 95 °C for 15 s, 60 °C for 60 s, and 95 °C for 30 s.

## 3. Statistical Analysis

Each experiment was conducted with at least three replicates. The mean differences were found to be statistically significant (*p* < 0.05) using one-way analysis of variance (ANOVA), which was performed using SPSS (version 20.0; SPSS Inc., Chicago, IL, USA), and independent sample *t*-tests, which were conducted with GraphPad Prism (version 8.0). To analyze the relative transcription levels of genes, the delta-delta Ct method was utilized. The qPCR amplification efficiency for each gene ranged from 95% to 105%, and the slope difference between the standard curves of all target genes and the internal reference gene was less than 0.1. Additionally, all statistical analyses were graphically represented using GraphPad Prism (version 8.0).

## 4. Results

### 4.1. PGPE Traits of Strain BHJ04

In vitro screening experiments revealed the following: strain BHJ04 could solubilize organic phosphate (Figure 1D) and inorganic phosphate (Figure 1E), fix nitrogen (Figure 1F), dissolve potassium (Figure 1C), produce siderophores (Figure 1B), and produce IAA (Figure 1G).

The effect of BHJ04 on the growth of *P. massoniana* seedlings was recorded (Figure 2F). Strain BHJ04 significantly increased in the number of fibrous roots of *P. massoniana*. Compared with the control, treatment with strain BHJ04 increased the diameter of the root by 36.05% (Figure 2B), the fresh weight of the aerial parts of the roots by 74.47% (Figure 2D), the fresh weight of the roots by 43.68% (Figure 2D), the dry weight of the aerial parts of the roots by 64.97% (Figure 2E), and the dry weight of the roots by 38.31% (Figure 2E). These data suggest that BHJ04 treatment significantly enhances *P. massoniana* seedling growth (*p* < 0.05).

### 4.2. Effect of BHJ04 on PWD in Pine Seedlings In Vivo

A seedling assay was used to analyze the effect of treatment with strain BHJ04. Treatment with BHJ04 significantly reduced the severity of the PWD (Figure 3). As PWD significantly decreased and disease severity significantly decreased compared with those in the control treatment, the following experiments were carried out.

### 4.3. Dynamic Expression of Several Genes in P. massoniana

In this study, we investigated the dynamic expression of genes involved in disease resistance in *P. massoniana*. After inoculation with strain BHJ04, the expression of genes encoding chitinase (Figure 4B), nicotinamide synthase (Figure 4A), and tetratricopeptide repeat-like superfamily protein isoform 9 (Figure 4D) was greater than that of the control samples in horsetail pine with pine wood nematode infestation at 6 h, 12 h, and 24 h. In the treatment group, the expression of the nicotinamide synthase and chitinase genes was greatest at 12 h, and the expression of the genes gradually decreased with time. The expression of the gene encoding the triangle-like tetrapeptide superfamily protein isoform 9 increased progressively with time, reaching a maximum at 24 h. In particular, the treated group showed much greater expression of chitinase, one of the most important disease-related proteins in plants. It was evident that inoculation with strain BHJ04 increased the expression of disease defense genes in infected *P. massoniana* and may also affect the regulation of their defense system against PWN infestation.

Plant cytochrome P450 family genes are essential in cellular detoxification mechanisms and are key components in the biosynthesis and evolution of metabolites with diverse biological activities in plants [37]. We investigated the changes in the expression of cytochrome P450 (Figure 4H) and its monooxygenase genes (Figure 4G) in treated and control *P. massoniana* trees. Compared with the control, inoculation with strain BHJ04 significantly enhanced the expression of cytochrome P450 and cytochrome P450 monooxygenase superfamily genes in infected *P. massoniana*. The expression of the gene encoding cytochrome P450 monooxygenase was downregulated at 24 h compared to 12 h after inoculation with the PWN. Nevertheless, the expression of the gene encoding cytochrome P450 gradually increased over time.

Pine wood nematode infestation affects the expression of genes associated with the replication of genetic material in *P. massoniana.* We found that the expression of DNA/RNA polymerase superfamily proteins (Figure 4C), genes related to DNA replication, was greater than that in the control samples at 6 h, 12 h, and 24 h after inoculation with strain BHJ04 in infected pine, and the expression gradually decreased. It was evident that inoculation with strain BHJ04 affects the process of replication of genetic material in pines vulnerable to PWN.

The dynamics of plant cell wall hydrolases respond to plant stress tolerance. The expression of the cell wall hydrolase gene (Figure 4F) was found to be much greater in samples treated with strain BHJ04 than in control samples, suggesting that inoculation with BHJ04 enhances the self-regulatory role of infested *P. massoniana* in defense against PWN infestation.

PWNs feed on tree epithelial cells and living thin-walled tissues, spreading and multiplying in the vascular system and resin pipes, ultimately obstructing water conduction [38]. The expression of the gene encoding a water stress-related gene, dehydration response protein (Figure 4E), was lower in BHJ04-treated pines infested with PWN at 6 h than in control samples. However, the expression of this gene was greater in the treatment group than in the control group at both 12 h and 24 h. Gene expression in the control samples gradually decreased over time, whereas gene expression in the treatment group increased and then decreased over the three periods. Thus, inoculation with BHJ04 temporarily increased the ability of infested ponytail pine to withstand water stress.

### 4.4. Colonisation of Roots, Stems, and Leaves of P. massoniana by Strain BHJ04

The EGFP-tagged strain BHJ04 was constructed to study its colonization of roots, branches, and leaves. Roots, stems, and leaves of *P. massoniana* in the experimental and control groups were sliced 30 days after inoculation with the EGFP-labelled strain BHJ04. The sections were placed on clean slides and observed with a laser scanning confocal microscope (LSCM). As shown in Figure 5, strain BHJ04 was distributed in the roots, vascular tissues of the stems, and needles of *P. massoniana*.

To determine the number of colonies in each period, absolute quantification was used, and the standard curve equation was Y = −3.508x + 40.106, R^2^ = 0.991, E = 97.795. 

There was an overall downwards trend in the number of BHJ04-labelled strains in roots, stems, and leaves during the test period. However, the number of strains was always greater than that in the control at 0 d. At the beginning of the experiment, strain BHJ04 numbers in roots and leaves decreased rapidly, then slowed after 45 d, and finally stabilized. The strains showed a more stable decreasing trend in the stems, and the final colonization after 60 d was greater than that in the roots and foliage. After 45 d of inoculation, the numbers of strain BHJ04 in *P. massoniana* roots, stems, and leaves were 4.56 × 10^3^ copies/μL, 9.73 × 10^3^ copies/μL and 2.16 × 10^3^ copies/μL, respectively. At 60 d, the amount of colonization in the tissue was lower than that at 0 d. Preliminary speculation that the gradual decrease in the numbers of strain BHJ04 in the horsetail pine in the body may be the cause of the gradual decrease in the numbers of marker bacteria in the roots, stems, and leaves in the internal colonization due to its gradual adaptation to the internal microenvironment of the pine seedling, resulting in many bacteria in the early stage of the death of many bacteria. After a period to adapt to the decrease in the number of bacteria, the bacterial strains gradually adapted to the conditions of internal survival of the tree and gradually reached a stable level (Figure 6). Overall, at the beginning of the experiment, the colonization of BHJ04 was greater in the foliage than in the roots of *P. massoniana*. In addition, the colonization in the roots was greater than that in the stems, but the greatest colonization was found in the stems at 60 d. The tendency to decrease in the branches was more stable than that in the roots and foliage, which may be attributed to the fact that this strain was isolated from the stems of *P. massoniana* and that the environment of the stems is more suitable for the survival and reproduction of the labelled strains.

## 5. Discussion

Our aim of this study was to elucidate the molecular mechanisms by which *P. abietaniphila* BHJ04 promotes growth and enhances disease resistance in *P. massoniana*. In our prior research, we identified 16S rRNA gene sequences of endophytic bacteria in healthy *P. massoniana* specimens collected from areas afflicted by nematodes, leading to the isolation of several cultivable bacterial strains. Strain BHJ04 is a strain with a high population proportion and potential for biocontrol (unpublished). We sequenced the whole genome of *P. abietaniphila* BHJ04, and this work has been extremely rare for this bacterium. Many genes related to plant growth promotion including those involved in nitrogen fixation, phosphorus solubilization, potassium solubilization, IAA production, and siderophore production have been identified in the genome. Strain BHJ04 was found to have the growth-promoting properties described above and could significantly promote the growth of pine branches and roots after intercropping with *P. massoniana*. Observations using LSCM and absolute qPCR analyses confirmed the colonization of BHJ04 on pine trees. Additionally, the results demonstrated that BHJ04 significantly curtailed the propagation of PWD.

Pine endophytic bacteria are believed to function in the interaction between pine plants and nematodes [39]. Research has demonstrated that PGPE in *Stenotrophomonas* sp., *Bacillus* sp. [40], *Streptomyces* sp. [41], and *Pseudomonas* sp. reduce the PWN population in pine trees, and these beneficial effects are attributed to the promotion of plant growth, the production of antinematode metabolites, and the induction of systemic resistance in the host plants [21,26,28]. Many effective PGPE biocontrol agents, such as the three bacterial strains *Pseudomonas putida* 16YSM-E48, *Curtobacterium pusillum* 16YSM-P180, and *Stenotrophomonas rhizophila* 16YSM-P39, which were isolated from pine trees by Kim, have been reported to induce systemic resistance. In our study, transcript levels were assessed not only for the defense-related gene chitinase, the stress-related gene nicotinamide synthetase, and tetratricopeptide repeat-like superfamily protein isoform 9 but also for the expression levels of the DNA replication-related genes DNA/RNA polymerase superfamily proteins, dehydration-responsive proteins, cell wall hydrolase-related genes, and the detoxification-related genes cytochrome P450 monooxygenase superfamily and cytochrome P450. The findings indicated that, upon inoculation with strain BHJ04, the expression of certain genes in *P. massoniana* increased in response to PWN invasion in comparison to that in the control group.

PGPE enhances the resistance of pine trees to PWN. This enhancement is frequently linked to the modulation of defense gene expression in pine trees [28]. Plant tissues respond to pathogen attack through a variety of biochemical reactions called defense responses. Pine treated with strain BHJ04 had higher relative transcript levels of chitinase; in the seedling experiment, seedlings treated with this strain had significantly less disease. Plant chitinases are thought to be closely related to plant pathogen defense and are PR proteins involved in defense against pathogen attack [25]. There is evidence that chitinase increases pine resistance to pathogens [24,42]. Chitinase expression induced by pathogen attack and injury has been demonstrated in *Pinus* [43,44]. Increased expression of chitinase-related genes and increased production of plant endochitinase may inhibit nematodes. Cytochrome P450 monooxygenases are among the most versatile and abundant plant biosynthetic enzymes. They are crucial in the detoxification mechanisms of cellular processes [45]. Throughout the evolution of the P450 enzyme family and its clades, there has been a notable shift in their biochemical functions. Initially, these enzymes were mainly involved in primary metabolism, including the biosynthesis of steroids and carotenoids. Over time, their function has expanded to secondary metabolism, particularly in the biosynthesis of defense molecules, such as alkaloids [46]. In the treated group, the expression of these genes consistently exceeded that of the control group as the duration of the infestation progressed. This pattern suggests an enhanced detoxification process in the pine, contributing to an increase in the plant’s systemic resistance. 

In terms of promoting tree growth, biocontrol microorganisms can reduce the severity and development of PWD by alleviating the typical symptoms of diseased trees, such as reducing photosynthesis, reducing water conductance, and disrupting transpiration [47]. In this study, we verified the nitrogen-fixing capability of strain BHJ04. The introduction of this nitrogen-fixing bacterium mitigated the loss of photosynthetic pigments and water in pine trees. Furthermore, it significantly increased the biosynthesis of soluble polyphenols and malondialdehyde in *Pinus sylvestris*, thereby enhancing the resistance of *P. sylvestris* to the PWN [48]. Moreover, strain BHJ04 solubilized phosphorus and potassium. Most minerals containing phosphorus and potassium are found in the soil in a fixed state, which makes them difficult for plants to use [49,50]. Furthermore, bacteria convert phosphorus and potassium fixed in the soil into active forms, which greatly increases the efficiency of nutrient uptake by plants [51]. Moreover, our findings indicated that strain BHJ04 can produce IAA, as evidenced by the Salkowski II reagent test. The introduction of IAA-producing bacteria has been shown to enhance the disease resistance of the host plants [52]. The introduction of phytohormone-producing bacteria into the root zone has been shown to enhance the biological control of nematodes, further contributing to the plant’s overall health and resistance to pests [53]. The phytohormones produced by plant growth-promoting rhizobacteria to promote plant growth and interaction with microorganisms may also explain this phenomenon [54]. Dramatic changes in the nematode communities associated with *Arabidopsis* root systems have been reported in almost all disruptions of genes involved in phytohormone pathways [55]. There is also a siderophore production capacity of strain BHJ04 in CAS media. Furthermore, metal carriers are known to be important weapons for bacteria to promote plant growth and interact with other microorganisms, nematodes, and higher animals and plants in the ecosystem [17]. Metabolites secreted by bacteria enriched in secondary metabolites, such as iron carriers, have potential for the biological control of pine wood nematodes under iron limitation [56]. The abundance and mobility of metals or metalloids in interrooted and surrounding environments are affected by metabolites such as siderophores, Fe^3+^, and Fe^2+^, which are critical for interactions among plants, microbes, and nematodes [57]. The characteristics of strain BHJ04 are crucial for promoting the growth of *P. massoniana* and may also be critical to improving its resistance to pine wilt disease.

The successful colonization of the pine trees by this strain is essential for controlling PWD, leveraging the intricate interactions between bacteria and pine trees to achieve disease management [58]. The genomic diversity observed at the species and strain level suggests that *Pseudomonas* spp. have the potential for a wide range of evolutionary adaptations to different environments [59]. Additionally, these organisms can control the proliferation of soil-borne pathogens [60], promote plant growth, and consequently contribute to an increase in crop yields [61]. We identified *P. abietaniphila* BHJ04 in the trunks of *P. massoniana*. Strain BHJ04 colonized the roots, stems, and leaves of *P. massoniana*, with initial colonization predominantly occurring in the leaves, albeit with limited longevity. Importantly, the colonization of strain BHJ04 on the trunks persisted for up to 60 d, exhibiting the slowest decline in viability. These findings suggested that strain BHJ04 holds promise as a biocontrol agent within the tissues of *P. massoniana* owing to its robust colonization capabilities.

## 6. Conclusions

Our results suggest that the selected bacterial strains may crucially enhance the growth and nutritional uptake of *P. massoniana* and potentially activate the plant’s innate immune defenses. To comprehensively elucidate the mechanisms through which these bacterial strains and their derived metabolites promote growth and enhance resistance, additional detailed research is essential. Such investigations could clarify the pathways involved and confirm the efficacy of these microbes, setting the stage for their application on a larger scale in the biocontrol of diseases affecting *P. massoniana*. This advancement could revolutionize the management strategies employed in forestry by integrating sustainable and biologically based solutions.

## Figures and Tables

**Figure 1 microorganisms-12-01089-f001:**
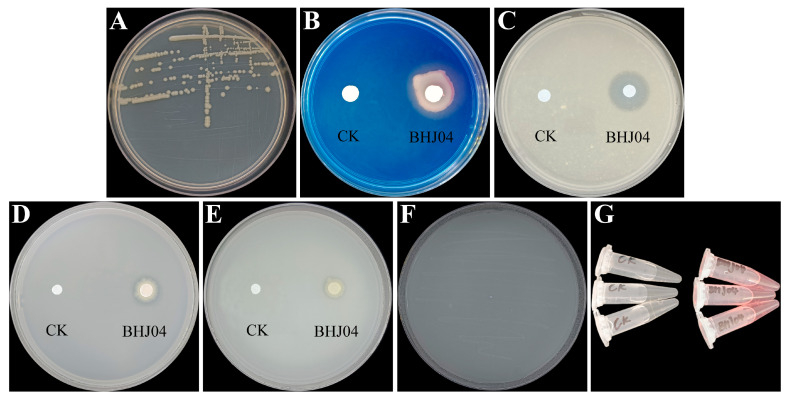
Plant growth promotion of strain BHJ04: (**A**) Colony morphology. (**B**) Siderophore production. (**C**) Potassium solubilization. (**D**) Organic phosphate solubilization. (**E**) Inorganic phosphate solubilization. (**F**) Nitrogen fixation. (**G**) Indole acetic acid production. LB is the control group.

**Figure 2 microorganisms-12-01089-f002:**
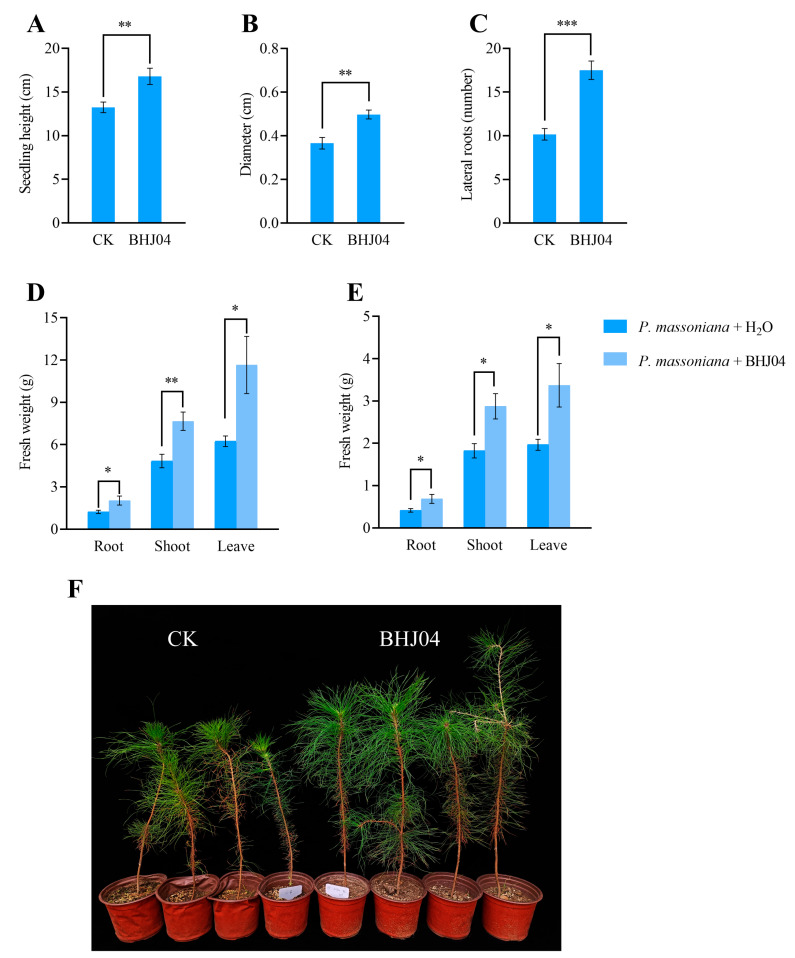
Growth promotion of *P. massoniana* seedlings by the BHJ04 bacterial suspension. (**A**) Seedling height. (**B**) Diameter. (**C**) Lateral roots. (**D**) Fresh weight. (**E**) Dry weight. (**F**) *P. massoniana* seedlings. LB medium was used as a control. Asterisks indicate significant differences according to Duncan’s multiple range test. “*” denotes *p* ≤ 0.05, “**” denotes *p* ≤ 0.01, “***” denotes *p* ≤ 0.001.

**Figure 3 microorganisms-12-01089-f003:**
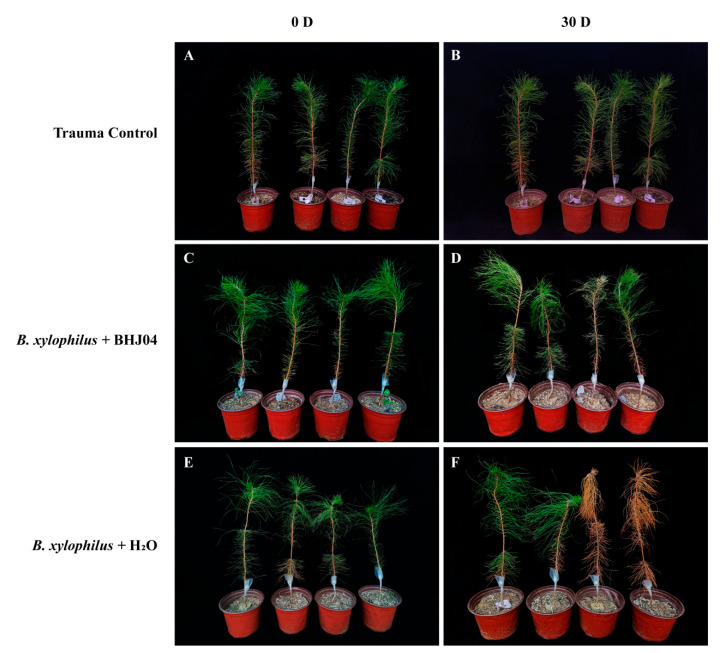
(**A**,**B**) Trauma control. Two-year-old *P. massoniana* plants were inoculated with a (**C**,**D**) BHJ04 bacterial suspension, inoculated with pine nematodes, or (**E**,**F**) inoculated with pine nematodes. Symptoms were relatively reduced in the BHJ04 bacterium suspension treatment group compared to the nematode inoculation group.

**Figure 4 microorganisms-12-01089-f004:**
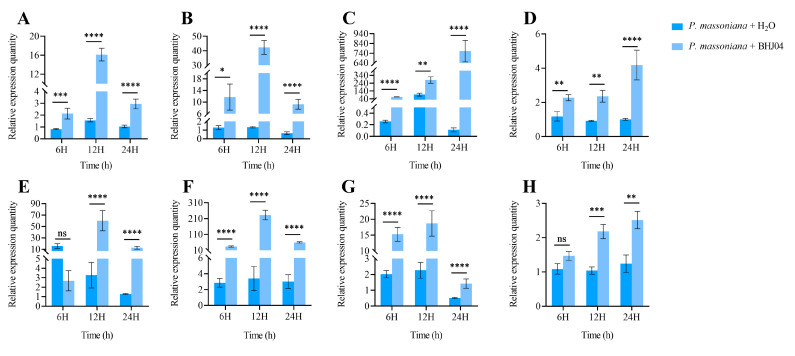
Relative transcript abundance of defense-related genes. (**A**) Nicotianamine synthase (NAS). (**B**) Chitinase. (**C**) DNA/RNA polymerase superfamily proteins. (**D**) Tetratricopeptide repeat-like superfamily protein isoform 9. (**E**) Dehydration-responsive proteins. (**F**) Cell wall-associated hydrolase. (**G**) Cytochrome P450 monooxygenase superfamily. (**H**) Cytochrome P450. An asterisk indicates a significant difference between the two treatments. In the data presented, “ns” indicates that the difference is not significant, “*” denotes *p* ≤ 0.05, “**” denotes *p* ≤ 0.01, “***” denotes *p* ≤ 0.001, and “****” denotes *p* ≤ 0.0001.

**Figure 5 microorganisms-12-01089-f005:**
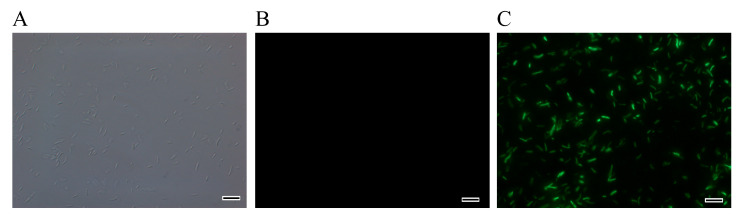
Morphology of strain BHJ04 observed under 100× LSCM. (**A**) Wild-type strain BHJ04. (**B**) EGFP-labelled strain BHJ04 in the dark field. (**C**) EGFP-labelled strain BHJ04. Scale: 10 μm.

**Figure 6 microorganisms-12-01089-f006:**
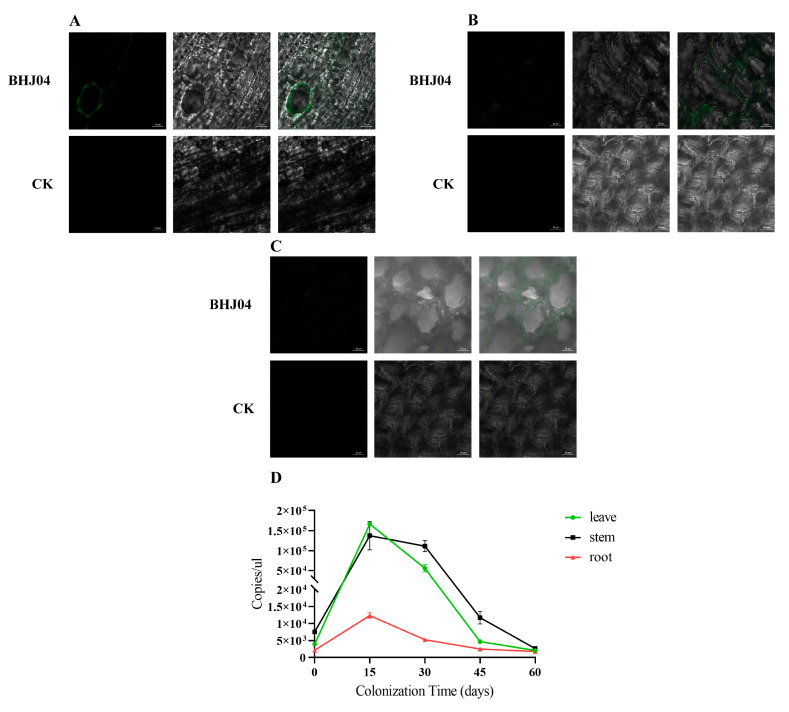
Detection of the fluorescent strain EGFP-BHJ04 colonization. (**A**) Colonization of strain EGFP-BHJ04 on pine leaves. (**B**) Colonization of strain EGFP-BHJ04 on pine stems. (**C**) Colonization of strain EGFP-BHJ04 on pine roots. From left to right. The first images were EGFP fluorescence-field bright-field images, the second images were bright-field images, and the third image was a combination of the first two images. Scale: 10 μm. (**D**) Colonization dynamics of *P. massoniana* strain BHJ04 inoculated for 0 d, 15 d, 30 d, 45 d, and 60 d as detected using RT-qPCR.

**Table 1 microorganisms-12-01089-t001:** Primer sequences of putative defense-related genes. NSA, chitinase, and tetratricopeptide repeat-like superfamily protein isoform 9 are related to defense genes in pine. DNA/RNA polymerase superfamily proteins, cell wall-associated hydrolases, and dehydration-responsive proteins are related to the stress response in pine. The cytochrome P450 and cytochrome P450 monooxygenase superfamilies are related to detoxification genes.

Gene	Primer Sequence (5′–3′)	F/R
Nicotianamine synthase (NAS)	TCCGCCGTCACCTTCT	F
	TTCGGTCCCGCCTAAT	R
Chitinase	CGAGGGCAAGGGATTCTA	F
	ATTCCTGGCTGTTGATGGC	R
DNA/RNA polymerases superfamily protein	TCAGTTCCGCTTATCACCCG	F
	CATCCGCACTTCGCTTCTC	R
Tetratricopeptide repeat-like superfamily protein isoform 9	ATGTGCTCATCGGGCTCT	F
	AGGGTGACTTGGCTTGT	R
Cytochrome P450 monooxygenase superfamily	AATCCGTCGTAGGCAACA	F
	GCCCGCCACATAGAAAT	R
Cytochrome P450	GTCGGAAACCTCCACCAAC	F
	TAGGGACTGAGCCCAAGC	R
Cell wall-associated hydrolase	CTCTAACCAAACTCCGAATACC	F
	CGCACTTCCGATACCTCCAT	R
Dehydration responsive protein	ATACTCATCTCGCCCACC	F
	GAGCGTTCTGTAAGCCTGT	R
Actin	CCTTGGCAATCCACATC	F
	TCACCACTACGGCAGAAC	R

## Data Availability

Data are contained within the article.

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
