# Peer review of "The Growth-Promoting and Colonization of the Pine Endophytic Pseudomonas abietaniphila for Pine Wilt Disease Control"

_microorganisms, 2024, doi:10.3390/microorganisms12061089_

Round 1

Reviewer 1 Report

Comments and Suggestions for Authors

The manuscript entitled: " Pine endophytic Pseudomonas abietaniphila with growth-promoting and colonization for pine wilt disease control. It is well-written and I'm pleased to recommend its acceptance for publication after major changes.

General Considerations:

The authors should carefully proof-read the entire manuscript to minimize typographical errors, especially with spellings, punctuations, unnecessary capitalizations, spaces, and units, as well as to ensure uniform expression of various special characters and abbreviations, terms, and phrases.

 Specific Considerations:

1.      The aim of study should be improved.

2.      Please the word in vitro should be written in italic in the whole manuscript.

3.      Please carefully revised the italics in the names of plants and bacteria species (also the genus).

4.      Please added a list of abbreviation at the end of the manuscript; (for example LB, IAA……)  

5.      Please added a concise conclusion into the paper.

6.      Please revise the language of the manuscript.

Comments on the Quality of English Language

The manuscript entitled: " Pine endophytic Pseudomonas abietaniphila with growth-promoting and colonization for pine wilt disease control. It is well-written and I'm pleased to recommend its acceptance for publication after major changes.

General Considerations:

The authors should carefully proof-read the entire manuscript to minimize typographical errors, especially with spellings, punctuations, unnecessary capitalizations, spaces, and units, as well as to ensure uniform expression of various special characters.

1.     

Reviewer 2 Report

Comments and Suggestions for Authors

This manuscript describes an experiment testing the ability of a bacterial strain, Pseudomonas abietaniphila BHJ04, to enhance Pinus massoniana resistance to Pine Wilt Disease (PWD) caused by nematodes. The testing included characterization of BHJ04 metabolism, co-inoculation with bacteria plus nematodes, and gene expression analysis in root, shoot, and leaf tissues.

The results of the experiment look promising for bacterial inoculation as an approach to combat PWD. However, the manuscript needs extensive English language editing in order to be acceptable for publication. Important method details are missing, or are not mentioned until the Results or Discussion sections. A few specific examples are given below.

The species of strain BHJ04 is not mentioned until section 3 of the Materials and Methods.

Line 71: It would be good to explain in the introduction why chitinase genes are relevant to nematode resistance, since often these genes are associated with defense against fungal pathogens.

The Materials and Methods section needs careful editing for consistency of tense. Right now it is partly in past tense, partly in present tense.

Lines 172-183: There is no mention in the Methods section of the length of time the trees were monitored and scored, nor the time points for observation or for RNA extraction.

Lines 222-227: It is not clear whether the nine Pseudomonas species’ genomes were sequenced as part of the current study, or whether they were obtained from an online database or a collaborator. It is also unclear what “covariance” means in terms of Mauve software – was it used to look for synteny, allelic variance, both?

Comments on the Quality of English Language

Extensive English language editing is needed.

Round 2

Reviewer 1 Report

Comments and Suggestions for Authors

the authors have satisfactory revised the manuscript according to the reviewer's comments. 

Author Response

Thank you very much for your positive feedback on the revisions made to the manuscript. I greatly appreciate your guidance and am pleased to hear that the changes have met your expectations. Thank you once again for your valuable input and support.